



# Reconstruction of past marine submersion events (storms and tsunamis) on the North Atlantic coast of Morocco

Otmane Khalfaoui[1,2], Laurent Dezileau[1], Jean-Philippe Degeai[3], Maria Snoussi[2]

[1]M2C - Morphodynamique Continentale et Côtière, UMR 6143, Université de Caen, 24 rue des tilleuls, 14000, Caen, France
[2]LGRN - Laboratoire de Géophysique et Risques Naturels, Centre de Recherche GEOPAC, Institue Scientifique, Université Mohammed V de Rabat; Avenue Ibn Battouta, B.P. 703 Agdal, 10090, Rabat, Morocco
[3]ASM - Archéologie des Sociétés Méditerranéennes, UMR 5140, Université de Montpellier3, CNRS, MCC, 34000, Montpellier, France

*Correspondence to*: Otmane Khalfaoui (m.otmanekhalfaoui@gmail.com) and Laurent Dezileau
(laurent.dezileau@unicaen.fr)

**Abstract.** The North Atlantic coast of Morocco has been affected historically by marine submersion events resulting from both storms and tsunamis and causing human and economic losses. The development of proactive adaptation strategies requires the study of these events over centennial to millennial timescales. Using a 2.7 m sediment core sampled from the Tahaddart estuary, we have been able to reconstruct past marine submersion events on this coastal area of Morocco over the
last 4000 years. The high-resolution sedimentological and geochemical analysis conducted on this core allows us to identify 14 sediment layers attributed to marine high-energy events. The core was dated with isotopic techniques ($^{137}$Cs, $^{210}$Pb$_{ex}$, $^{14}$C) and the outcomes reveal that three sediment layers are in connection with two major historical marine submersion events. The first layer mentioned as E1 seems to fit with the great Lisbon tsunami in 1755 CE (Common Era), an event dated for the first time on the Atlantic coast of Morocco. The other two layers referred as E13 and E14 were dated between 3464 and 2837
cal BP and correlated with marine submersion deposits found on Spanish and Moroccan coasts, which confirms the existence of a major high-energy event (around 3200 BP) similar to the one in 1755 CE.

## 1 Introduction

Marine submersions are temporary floodings of the coastal zones due to storms or tsunamis (Chaumillon et al., 2017). With the increase of population and infrastructure in coastal areas, these events can cause significant human and economic losses
(Neumann et al., 2015). We can mention some events that have marked the world population by the huge damage they caused, like the two tsunamis of Sumatra in 2004 and Japan in 2011, or the cyclones of Katrina in 2005 and Haiyan in 2013 (Mori et al., 2014; Titov, 2005). Studies on past and present marine submersions are obvious for countries threatened by this coastal hazard, in order to develop strategies for coastal communities protection and to mitigate as much as possible future damages (Alcántara-Ayala, 2002).

The North Atlantic coast of Morocco has been affected by numerous marine submersions (Mellas, 2012; Snoussi et al., 2008). This coastal area has experienced extreme marine storms due to the effects of the North Atlantic Oscillation (NAO) (El Messaoudi et al., 2015). Moreover, this area is also subject to tsunami events, which can come from two main sources: (1) a seismic source related to the tectonic activities in the Gulf of Cadiz and Alboran Sea. Indeed this area is located not far from a convergent boundary between African and Eurasian plates. The African plate is diving below the Eurasian plate,
causing earthquakes, some of which have been able to generate tsunami waves (e.g. the 1755 CE (Common era) Lisbon tsunami) (El Alami and Tinti, 1991; Gutscher et al., 2006); (2) a volcanic source, that has been highly publicized, is the Cumbre Vieja volcano located in the Canary Islands. A large eruption from this volcano could cause a flank slide, leading to a large amplitude tsunami (Ward and Day, 2001).

The current textual archive and instrumental data on storms and tsunamis in Morocco do not allow us to deduce any
evolution in time, due to the lack of data covering a sufficiently long time period (Raji et al., 2015). Since these events are





causing sedimentary inputs on the littoral domain, the geological archives present in this area will permit the study of these events over a longer period of time, beyond the instrumental and textural records (Degeai et al., 2015; Dezileau et al., 2011; Morton et al., 2007; Sabatier et al., 2012). This geological approach has been used on the Portuguese and Spanish coasts (Costa et al., 2012; Dawson et al., 1995; Font et al., 2013; Hindson et al., 1998; Luque et al., 2001; Morales et al., 2008).

Conversely, only a few studies have been conducted on the Moroccan Atlantic coast (Chahid et al., 2016; Leorri et al., 2010; Mhammdi et al., 2015; El Talibi et al., 2016). In this context, the present work aims to reconstruct past marine submersion events from geological archives (cores) collected in Tahaddart estuary (NW of Morocco) using a high-resolution sedimentological and geochemical analysis.

## 2 Study area

Tahaddart Estuary is located in the northern part of the Moroccan Atlantic coast, between Tangier city and Assilah city (coordinates: 35°30' and 35°40' latitudes North and 5°55' and 6°01' longitudes West). The estuary is a protected area, it was classified in 1996 as a biological and ecological site of interest (SIBE) and in 2005 as a RAMSAR area (The Convention on Wetlands) (Fig. 1).

Tahaddart River watershed has an area of approximately 1,200 km$^2$. The hydrographic network is formed by two main
entities separated by Haouta Beni Mediar hill formation: the M'Harhar River on the northern part and the Hachef River on the southern part. These two rivers meet 4 km from the Atlantic coast to form the Tahaddart River. The Tahaddart watershed is located on the NW part of the Rif chain; it is formed by the dominance of marly clay and marly limestone deposits of Meso-Cenozoic age. The Quaternary is mainly represented by fluvial terraces and glaciers of continental accumulations (Durand-Delga and Kornprobst, 1985; Suter, 1980).

The region has a Mediterranean climate tempered by Atlantic influences (El Gharbaoui, 1981). It is a very windy region (with a maximum speed of 130 km/h.), exposed to western winds from December to April, and to eastern winds from March to November (Jaaidi et al., 1993; Taaouati et al., 2015). The area is classified as mesotidal, with a range of 2.7 m and a semidiurnal periodicity (ONE, 2002). The frequent wave heights are between 0.5 and 1m (around 40%), the heights between 1 and 1.5 m are less frequent (around 20%). The periods of these swells range from 3 to 16 seconds (Nachite et al., 2010).

The longshore drift dominant direction is from North to South caused by west-northwest swells (Achab, 2011).

**Figure 1. Location of the Tahaddart estuary and the TAH17-1 coring site.**

## 3 Sampling and analytical methods

A sedimentary core with 2.7 m in length and 5 cm in diameter (TAH17-1) was extracted from Tahaddart estuary in
5   September 2017 with the following geographical coordinates: 35°34'30.94"N, 5°59'7.79"W (Fig. 1). The distance between
the coring site and the shoreline is approximately 1 km. The coring operation was undertaken using a "Cobra" portable
percussion corer and a hydraulic puller (Géosciences Montpellier Laboratory, France). Once in the laboratory, the core was
cut, photographed and then transported to a cold room to prevent desiccation of the sediment. In addition, 39 surface
sediment samples were collected from the high Tahaddart River watershed to the littoral area (beaches and dunes) in order to
10   identify all sources of sediment arriving in the study area (Fig. 2).





### 3.1 Grain size

The particle size distributions were determined using a Beckman-Coulter LS13320 laser diffraction particle size analyser (Géosciences Montpellier Laboratory, France). The analysis was carried out continuously along the core (each centimetre) and only on the particles lower than 2 mm in diameter. Samples were suspended in deionized water before measurement and exposed to ultrasound during the measurement to prevent flocculation. The particle sizes obtained are classified according to Folk and Ward (1957).

### 3.2 Geochemistry

Geochemical analysis by X-ray fluorescence (XRF) spectrometry was performed every centimetre on the surface of the half-core TAH17-1 using a hand-held Niton XL3t spectrometer (pXRF; Geoscience Montpellier Laboratory, France). The half-core was first covered by a thin layer of ultralene film to avoid contamination when changing the measuring point (Richter et al., 2006).

### 3.3 Geochronology ($^{210}Pb_{ex}$, $^{137}Cs$ and $^{14}C$)

Radiochronological data for the last century was obtained using $^{137}Cs$ and $^{210}Pb_{ex}$ measurements with a CANBERRA BEGe 3825 gamma spectrometer (Géoscience Montpellier Laboratory, France). The $^{137}Cs$ activity vs. depth distributions was done according to Robbins and Edgington (1975) while CFCS ("constant flux, constant sedimentation rate") model was used for the $^{210}Pb_{ex}$ (Golberg, 1963; Krishnaswamy et al., 1971). In order to complete the chronology over longer periods of time, $^{14}C$ dates were performed on 5 marine shells in the LMC14 laboratory (the ARTEMIS accelerator mass spectrometry at the CEA institute in Saclay (Atomic Energy Commission)) according to the procedure described by Tisnérat-Laborde et al., (2001). The calibration of $^{14}C$ dates was done using the CALIB 7.1 program (Stuiver et al., 2017).

## 4 Results and discussion

### 4.1 Lithostratigraphy description

The lithological description was based on visual observation of changes in sediment texture, colour and sedimentation structures. Three lithological units have been distinguished and presented with a schematic log (Fig. 2):

Unit A (140-270 cm depth): the bottommost unit formed by an alternation of fine (silt) and coarse (sand) sediments. However, from the base to the top of the unit, there is a progressive change in the colour of the sediment from brown to dark grey. Two subunits can be distinguished:

- Subunit A1 (195-270 cm depth): this subunit is 0.75 m thick, characterised by a dominance of brown silty sand with some marine shell fragments. An enrichment with coarse sand was observed between 240 and 260 cm depth with the presence of a rip-up clast (3x4 cm in size).
- Subunit A2 (140-195 cm depth): with a thickness of 0.55 m, this subunit is formed mainly by a dominance of grey to dark silt with several distinct layers of grey sand (marked by sandy peaks up to 90%) (Fig. 2). The grey to black colour of this subunit is due to the presence of organic matter in the form of plant remains (roots).

The lithological characteristics of these subunits indicate most likely a low-energy environment, dominated by silt to silty sand sediments but interrupted by injections of sandy layers, who arrive at the coring site probably by high-energy processes (e.g. storms or tsunamis).

Unit B (85-140 cm depth): The unit is about 0.20 m thick, consists of chaotic and structureless sediments, ranging from silt to pebbles (especially at the base). The contact with the underlying unit is discontinuous and marked by the presence of a grey rip-up clast (2.5x3.5cm in size), the colour of the clast suggest a provenance by erosion from subunit A1. This unit





seems to share common properties with some tsunami deposits in the Portugal coast (unit E in Costa et al., (2012); unit B in Dawson et al., (1995)).

Unit C (0-85 cm depth): The topmost unit presenting a thickness of 0.85 m and marked by a massive brown to grey silt with traces of oxidized plant roots between 30 and 85 cm depth.

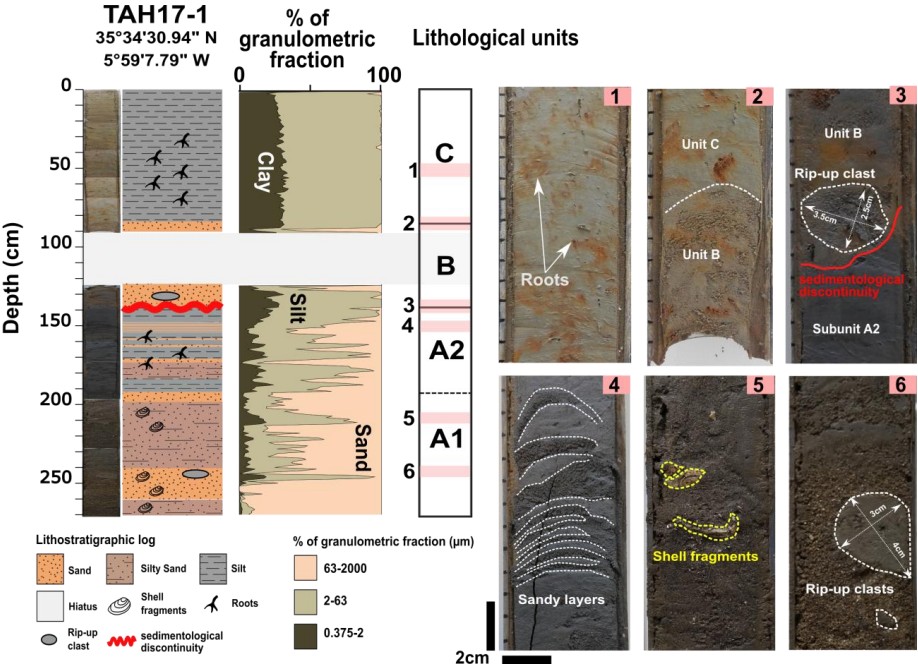

**Figure 2. Simplified sedimentological log showing the principal lithological units of the TAH17-1 core.**

### 4.2 Geochemistry

#### 4.2.1 Source of sediments

A first step in the identification of past marine submersion deposits is to determine the sources of sediments arriving to the study area (Dezileau et al., 2011). A total of 39 surface samples were collected, including 19 from the Tahaddart River watershed and 20 from the coastal area (beaches and dunes). These samples were analysed by pXRF and 16 chemical elements were detected with different concentration: Si, Rb, K, Cr, V, Zn, Pb, Fe, Mn, Ni, Zr, Ca, Sr, Ti, Cu and As. A principal component analysis (PCA) was performed. The two factors of the PCA explain 62.20 and 19.10% of the variance (Fig. 3a). The results show two geochemical poles: a first pole gathering elements like Rb, K, Cr, V, Zn, Pb, Fe, Mn, Ni, Ti and Cu, and a second pole gathering elements like Ca, Sr, As and Zr.

Concentration maps of Rb and Ca (representing each chemical pole) are presented in Fig. 3b-c. Rb concentrations are high in samples from the watershed and low in coastal samples, which indicates that elements of the first geochemical pole (Rb, K, Cr, V, Zn, Pb, Fe, Mn, Ni, Ti and Cu) represent better sediments arriving from the watershed. On the other hand, Ca concentration is high in coastal samples and low in watershed samples, which make elements of the second geochemical pole (Ca, Sr, As and Zr) characterise the marine sediments. Silicon is not included in one of the two groups, because in our case, it can come from two sources (continental and marine).

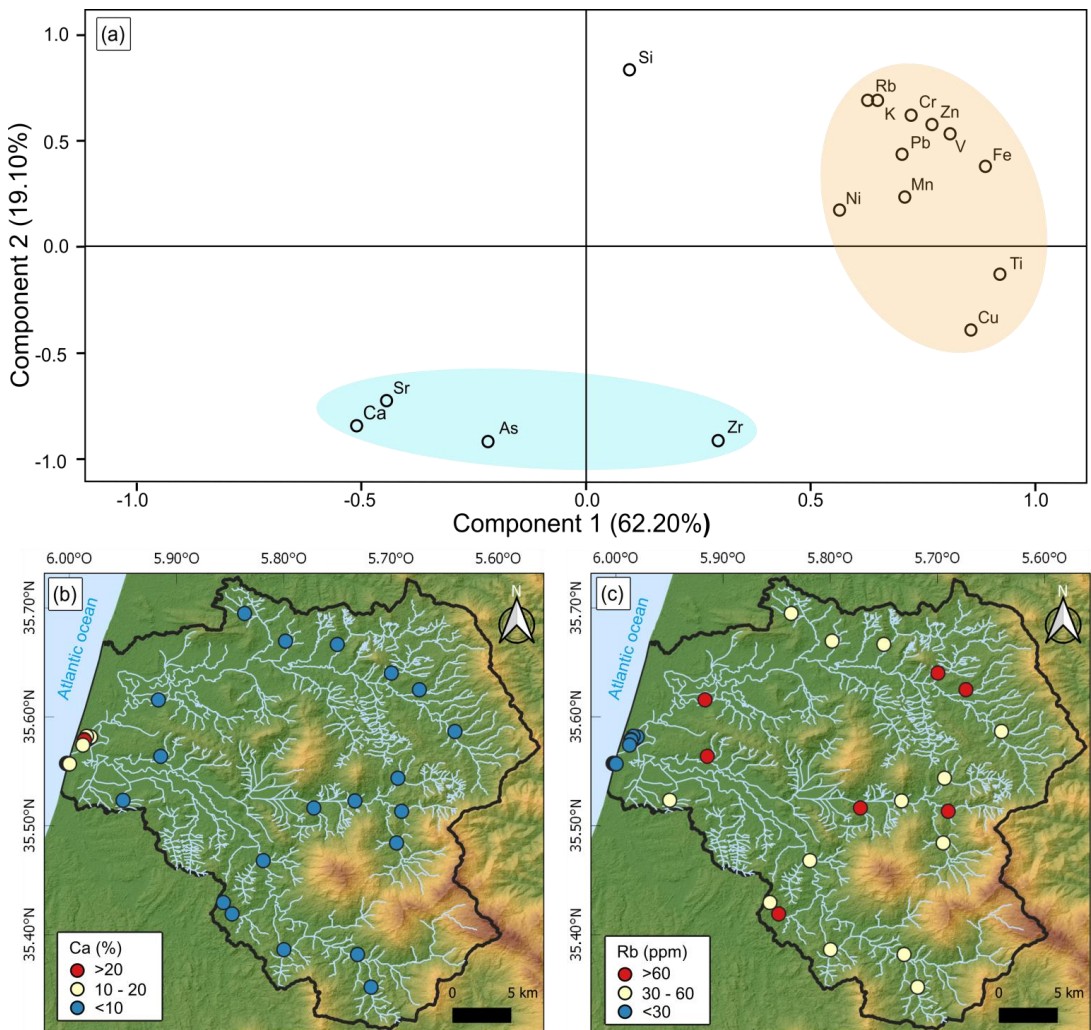

**Figure 3. (a) Principal Component Analysis (PCA) of the geochemical elements obtained in the surface samples. (b),(c) Concentration maps of Ca and Rb in surface samples taken from the coastal area and Tahaddart River watershed.**

### 4.2.2 Geochemistry of TAH17-1 core

5    In total, 25 chemical elements (Al, Si, P, S, Cl, K, Ca, Ti, V, Cr, Mn, Fe, Ni, Cu, Zn, Ga, As, Br, Rb, Sr, Y, Zr, La, Ta, Bi) were detected in TAH17-1 core using pXRF. Only elements that show significant variation with depth are shown in Fig. 4. Zr and Ca represent the marine geochemical pole, while Rb and Fe represent the terrigenous one. The concentrations indicate that Rb and Fe are associated with the silt-clay fraction. Conversely, the Ca and Zr are associated with the sandy fraction. These results confirm those obtained by the PCA in surface samples.


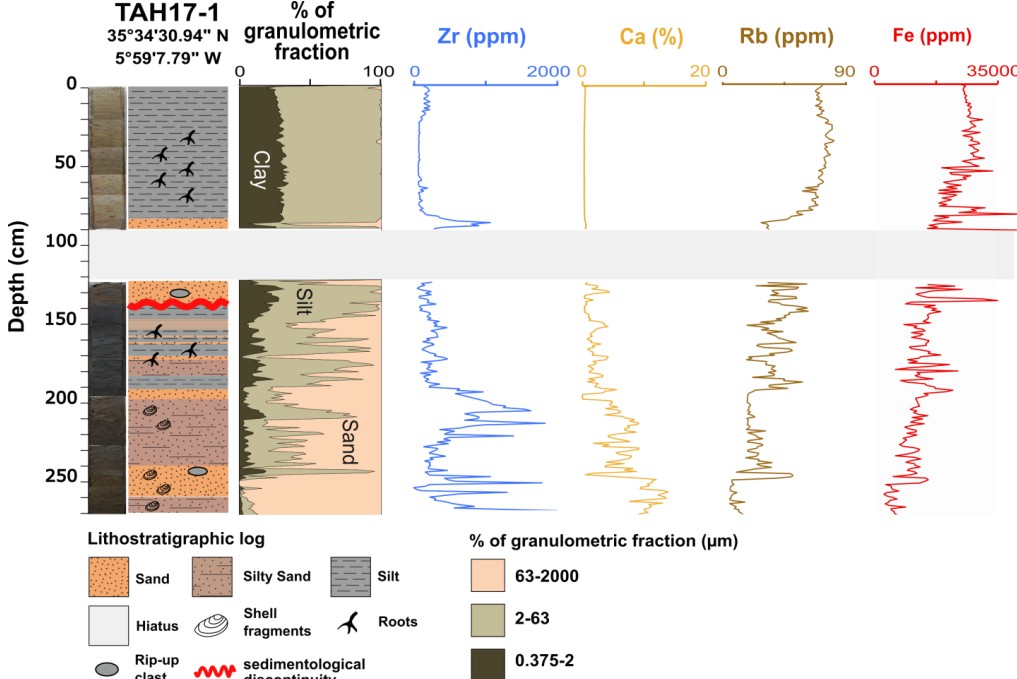

**Figure 4. Down-core plots of signal intensity vs. depth in TAH17-1 of Zr, Ca, Rb and Fe.**

## 4.3 Identification and chronology of marine submersion events

Based on our sedimentological and geochemical results, 14 sediment layers attributed to marine submersion events have been distinguished in TAH17-1 core (Fig. 5). These layers present at least one of the following characteristics:

- a well-defined sandy layer, interbedded in fine estuarine sediments;
- an enrichment in Ca and Zr indicating a marine source of the sediment, and depletion in terrigenous elements (Rb);
- the presence of sedimentation structures such as rip-up clasts and/or sediment discontinuity.

Among these identified sedimentary layers, three of them (E1, E13 and E14) have been dated using different geochronological approaches ($^{137}$Cs, $^{210}$Pb$_{ex}$ and $^{14}$C). For the first layer (E1), the presence of a homogeneous deposit between the top of the core and the top of the E1 layer (0-85 cm depth) indicates probably a constant rate of sedimentation. Using the accumulation rates calculated with the $^{137}$Cs and $^{210}$Pb$_{ex}$, we can deduce an extrapolated age for E1. The sedimentation rate estimated by the $^{137}$Cs, gives a value around 0.31cm/year and by the $^{210}$Pb$_{ex}$ method a value around 0.46cm/year (Fig. 5). The extrapolated age for E1 corresponds to 1742 +/-30 CE with the $^{137}$Cs and 1829 +/-30 CE for the $^{210}$Pb$_{ex}$, which give 1785 CE as an average between the two dates.

The two other layers (E13 and E14) were dated by $^{14}$C geochronology. Five shells sampled from the TAH17-1 core were dated using conventional Accelerator Mass Spectrometry (AMS) $^{14}$C measurements, and the radiocarbon ages were calibrated using the Marine13 curve (Reimer et al., 2013). To calculate the marine reservoir age R(t) and its regional deviation ΔR, two additional samples composed with a marine shell and a piece of wood (referenced as SacA 54447 and SacA 54446, respectively) were dated with $^{14}$C technique. Both samples were taken from another core (TAH17-4) near to our coring site. Since these two samples were found at the same depth, we suppose they have the same age of deposition, but the $^{14}$C value obtained for the marine shell (i.e. 1105 ± 30 BP) is older than the piece of wood (i.e. 810 ± 30 BP). This difference is due to the effect of the marine reservoir age R(t), which corresponds to 295 years in our case. Based on the Intcal13 curve (Reimer et al., 2013), we can estimate a historical age for the marine shell using the atmospheric $^{14}$C value of the wood (1240 CE; Reimer et al., 2013). The ΔR value is then calculated by subtracting the marine model age value





estimated at 1240 CE (1180 ± 24 $^{14}$C years; Reimer et al., 2013) from the measured apparent $^{14}$C age of the marine shell (1105 ± 30 BP years; Table 1), which corresponds to -75 ± 20 years. The calibrated $^{14}$C ages for the five shells taking into account the marine reservoir age R (t) and its ΔR are presented in Table 2.

5 **Table 1. Estimation of marine reservoir age R(t) and its regional deviation ΔR using $^{14}$C ages on wood and marine shell.**

| Lab code | Material dated | $^{14}$C age (BP) | Historical age using Intcal13 curve | Marine reservoir age R(t) (years) | Model age (Marine13 curve) | ΔR (Years) |
|---|---|---|---|---|---|---|
| SacA 54447 | Marine shell | 1105 ±30 | | 295 | 1180 ±24 | -75 ±20 |
| SacA 54446 | wood | 810 ±30 | 1240 CE | | | |

**Table 2. Calibrated $^{14}$C ages of samples taken from the TAH17-1 core.**

| Lab code | Depths (cm) | Dated material | δ13C (‰) | $^{14}$C age (BP) | $^{14}$C age (cal BP) (two sigma ranges) [start:end] |
|---|---|---|---|---|---|
| SacA 54435 | 214 | Marine shell | -1,90 | 3090±30 | [cal BP 2837: cal BP 3072] |
| SacA 54436 | 225 | Marine shell | -2,30 | 3195±30 | [cal BP 2970: cal BP 3209] |
| SacA 54433 | 251 | Marine shell | 4,10 | 3415±30 | [cal BP 3261: cal BP 3464] |
| SacA 54437 | 261 | Marine shell | -2,40 | 4105±30 | [cal BP 4144: cal BP 4388]* |
| SacA 54434 | 264 | Marine shell | 3,50 | 3570±30 | [cal BP 3441: cal BP 3648] |

*Age inversion.
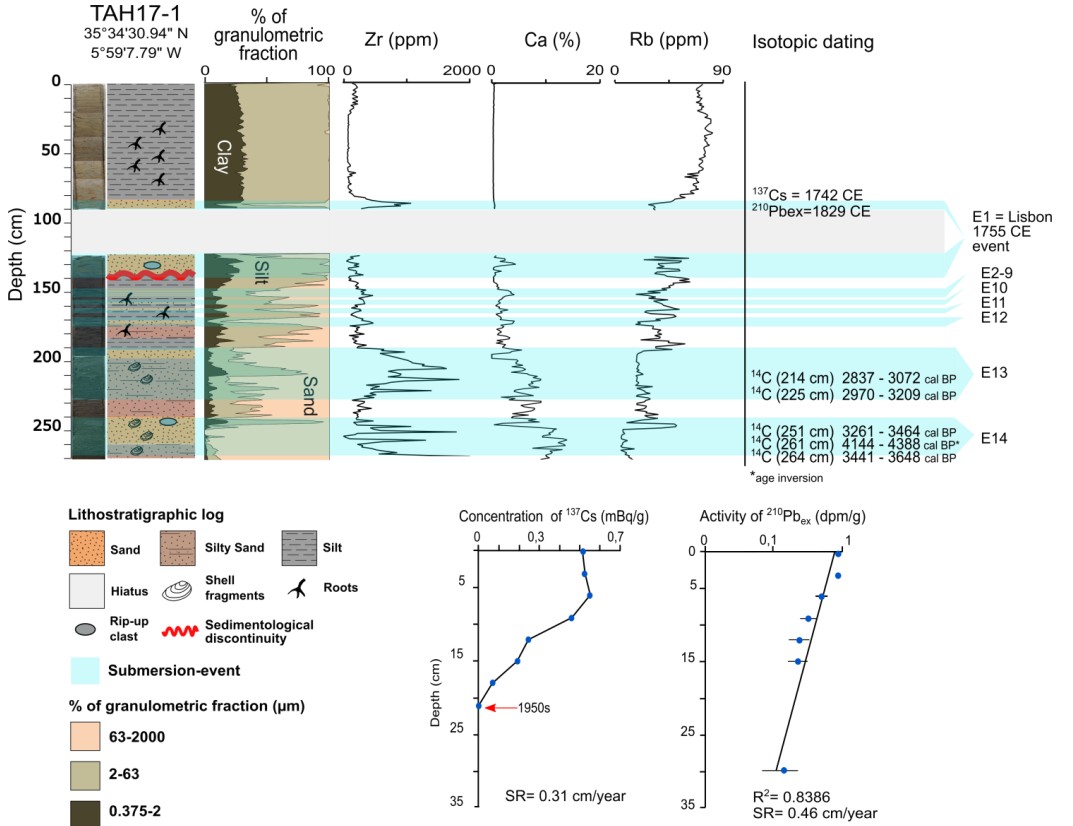

**Figure 5. Identification of marine submersion events in the TAH17-1 core and isotopic dating of E1, E13 and E14. SR: Sedimentation Rate.**

**4.4 Correlation of dated events with other historical archives**

A correlation was made between dated events in this study, with historical marine submersion events in the region. For E1, the extrapolated dates obtained by the $^{137}$Cs and $^{210}$Pb$_{ex}$ methods are in line with the great Lisbon tsunami in 1755 CE, a historical event considered as exceptional (Blanc, 2011; Kaabouben et al., 2009). From a sedimentological point of view, several deposits similar to E1 have been dated and attributed to the 1755 Lisbon tsunami on the Spanish and Portuguese coasts (Costa et al., 2012; Dawson et al., 1995; Luque et al., 2001; Morales et al., 2008). On the Moroccan coasts, some

sedimentary deposits have been indirectly attributed to this event without any high precision geochronology (El Talibi et al., 2016). In this study, the multi-dating approach ($^{137}$Cs, $^{210}$Pb$_{ex}$ and $^{14}$C) permits to associate for the first time, a sedimentary deposit (E1) to the 1755 CE Lisbon tsunami in the Moroccan coast.

The $^{14}$C dates obtained for E13 (3209-2837 cal BP) and E14 (3464-3261 cal BP) are close since we have a large uncertainty due to calibration procedures. We can assume that they could be multiple deposition layers for a single major event around

3200 BP (called in this case E13/14). Considering the thickness of the layer (E13+E14) and the presence of rip-up clasts at its base (similar to the one in E1), we can possibly consider this layer as tsunami deposit. Additionally, studies carried out on the Spanish coasts have shown the presence of high-energy marine deposits with ages close to 3200 BP (Fig. 6) (Lario et al., 2011; Ruiz et al., 2013), which confirm at the regional scale the existence of a marine submersion event during this period. On the Moroccan Atlantic coast, this event (E13/14) could be contemporaneous to marine deposits identified by Mhammdi

et al., (2015) on the Loukkos estuary with an estimated mid-Holocene age, and by Chahid et al., (2016) on the Rabat-Skhirat coast with an age between 2700 and 2200 cal BP.


For the other identified marine layers (from E2 to E12), we were unable to give an age to them due to the lack of $^{14}$C datable material. A combination of radiocarbon and optical dating techniques can be more efficient since the latter has been applied successfully to date tsunami deposits on the Spanish and Portuguese coasts (Dawson et al., 1995; Koster and Reicherter, 2014). For the moment, we consider these layers as separate marine submersion events; we cannot indicate whether they are 5 storms or tsunamis. However, the results presented in this study can be useful when undertaking further investigation in the area.

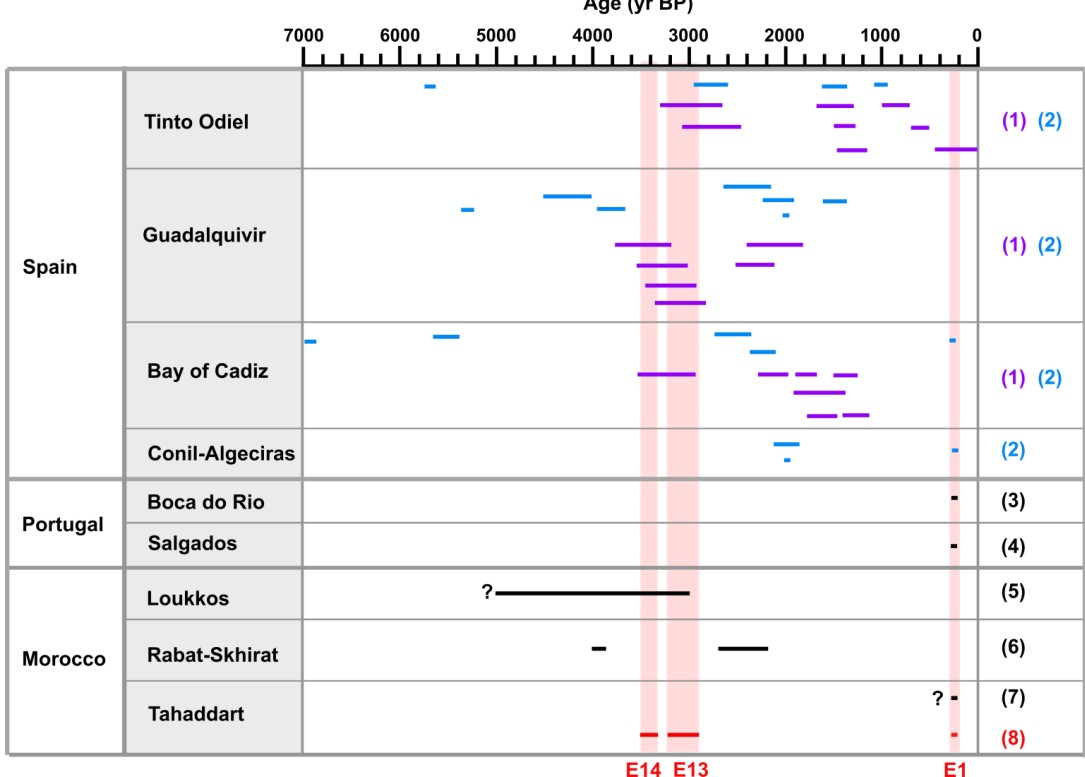

Figure 6. Correlation of dated events with other on-shore deposits in Spain, Portugal and Morocco. (1) Tsunamigenic deposits summarised by Ruiz et al., (2013). (2) Tsunamigenic deposits summarised by Lario et al., (2011). (3) (Dawson et al., 1995). (4) 10 (Costa et al., 2012). (5) (Mhammdi et al., 2015). (6) (Chahid et al., 2016). (7) (El Talibi et al., 2016). (8) This study (Khalfaoui et al.,).

## 5 Conclusion

Our study presents a 4000 years record of past marine submersion events using high-resolution multi-proxy analysis on a sediment core retrieved from Tahaddart estuary (NW of Morocco). The major finding in this study is the identification of 15 sedimentary imprints of two historical marine submersion events. The first one was the great Lisbon tsunami in 1755 CE that was related to E1 deposit. The second event was highlighted by E13 and E14 deposits, seems to correspond to another historical event identified on the Spanish and Moroccan coasts around 3200 BP. Further investigation is required using cross-shore and long-shore cores, coupled with more proxies (e.g. Anisotropy of Magnetic Susceptibility (AMS) and microfaunal analysis) to provide more information about these events, particularly the other deposits (From E2 to E12). 20 However, the results presented in this study will definitely feed the regional database on past extreme sea events, especially for the Moroccan Atlantic coasts, where there is a lack of studies on this topic. The objective is to assist as much as possible coastal managers to develop proactive adaptation strategies and protect the coastal population from damages that can be caused by these events.



## Competing interests

The authors declare that they have no conflict of interest.

## Acknowledgements

The authors would like to thank the Laboratoire de Mesure [14]C (LMC14) ARTEMIS at the CEA Institute at Saclay (French

Atomic Energy Commission) for the [14]C analyses. This study was funded by MISTRALS/PALEOMEX project and the
Partenariat Hubert Curien (PHC) Toubkal (No. TBK/17/40 - Campus No. 36864YB; coordinated by Laurent Dezileau and
Maria Snoussi).

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
