# Peer review of "Reconstruction of past marine submersion events (storms and tsunamis) on the North Atlantic coast of Morocco"

_Natural Hazards and Earth System Sciences, 2019_

## Referee Comment (RC1) · Raphael Paris (Referee) · 28 Jun 2019

Dear colleagues,

The sediment core that is presented in your paper represents the most complete sequence of marine submersion events ever found on the coasts of Morocco, including a good candidate for the famous 1755 tsunami. However, the paper has several major weaknesses and I regret to recommend its rejection. As it stands, the paper is rather a preliminary report on one single core, and more investigation is required before this study could be accepted for publication.

The title itself does not reflect exactly the content of the paper, as the distinction between storm and tsunami beds is not discussed in the paper. The potential storm deposits are not (or very-poorly) described.

In the introduction, more references are needed (see also my comment below). Previous references on marine submersion events on the coasts of Morocco are cited, but it would be good to add a short paragraph summarising the main conclusions of these papers.

I was surprised that only one core (TAH17-1) is presented, although at least one other core (TAH17-4, mentioned at line 20 page 7) was retrieved. Presenting other cores would allow evaluating the reproducibility of the results obtained for TAH17-1, together with giving information on the longitudinal/lateral continuity of the sedimentary units observed. Indeed, many tsunami deposits are characterised by longitudinal trends such as landward fining and thinning. Making conclusions from one core only is a strong limitation.

Methods: the geochemical analysis is based on data provided by a hand-held XRF device. The reliability of such equipment is questionable. As a first approach it gives a good idea of the composition of the sediment, but why not using a $\mu$-XRF lab device since you have good cores?

Why merging results and discussion in section 4? Sections 4.1 and 4.2. are clearly "result" parts. Discussion really starts at page 7 (section 4.3). Please consider reorganising the manuscript so as to separate results and discussion.

Description of the lithostratigraphy is too short and should include a description of the grain size distribution of individual subunits, and vertical trends of grain size.

On figure 2, there's a sedimentary gap at depth 90-120 cm. It is thus difficult to have a good idea of the characteristics of unit B, which is a key unit, probably related to the 1755 tsunami. Is it due to sand percolation during core retrieval? In that case why not

trying other cores? Why not trenching in order to access to this important unit? It's another severe limitation for the conclusions that are proposed.

At the beginning of page 5, unit B is compared to other 1755 tsunami deposits in Portugal. This comparison could be enlarged to many other tsunami deposits in the world (i.e. marine sand beds as discontinuities in finer sediments), but the sedimentary setting in Portugal and Morocco is certainly different, so more arguments are needed.

The XRF profiles seen on figure 4 should be described in more details (actually the section presenting these results is only 5-lines long).

At page 9 (correlation of events with historical archives), we lack a discussion on the magnitude/intensity of the 1755 tsunami on the Moroccan coast. There's indeed a great debate on the impact of this event and the Moroccan piece of the puzzle is particularly important for deciphering between the different rupture scenarios that are proposed in the litterature (see papers by MA Baptista, PL Blanc, etc). The tsunami evidence at 1 km from the shoreline provided by your new data is a key observations, but it is (too) briefly presented and discussed, and so under-exploited.

In the description of the core, you distinguished E13 and E14 subunits, but suddenly in the discussion these 2 subunits become a single tsunami deposit because it fits better with ages! A better description of the E13/E14 section of the core is required before concluding if it's one or 2 different events.

By the way, English should be revised by a native.

References:

More references in the introduction would be welcome (e.g. at lines 23, 27, 34, 36). Please avoid auto-citation when you cite a single paper illustrating a large topic (e.g. at line 23: Chaumillon et al. 2017 deals with storms only, although it's an excellent review).

Minor comments:

p1-l33: not sure if a tsunami from Alboran Sea would impact the Atlantic coast of Morocco. p1-l34: plunging (not diving). p1-l36-38: the Cumbre Vieja case viewed from Ward & Day (2001) is highly controversial. Please add other references on this low-probability / high-magnitude scenario (e.g. Abadie et al., 2012). p2-l3: "This approach has been used on the Portuguese and Spanish coasts, especially to track past tsunamis." p4-l36: I don't understand why unit B is 20 cm thick between 140 and 85 cm depth... p5-l1-2: this sentence should be moved to discussion. p5-l9-12: this section belongs to "methods". p5-l20: silica (rather than silicone). p7: it would be good to remind the depth range of subunits E1, E13 and E14 in the text. p7-l15: I'm not familiar with the use of such extrapolated ages, but is it really correct to present such an averaged date from two different techniques? p7-l23: The reservoir effect is not the only possible explanation for the difference observed between C-14 age on shell and wood. Where does they both come from in such a sedimentary setting? no transport prior to their present-day position? p10-l13: I would not write "high-resolution". High-resolution studies rely on other techniques such as $\mu$-XRF, CT-scan, etc.

Figures:

Figures 4 and 5 could be merged together. Place subunits E1, E13 and E14 on figures 2 and 4.

---

## Author Comment (AC1) · 1 Aug 2019

The authors would like to thank Professor Raphaël Paris for his valuable comments and suggestions, they will be seriously taking into consideration and corresponding corrections will be made in the next version of the manuscript. However, we present some clarification and answers (R) to his questions (Q) in the following text:

Q1- Dear colleagues, The sediment core that is presented in your paper represents the most complete sequence of marine submersion events ever found on the coasts of Morocco, including a good candidate for the famous 1755 tsunami. However, the paper has several major weaknesses and I regret to recommend its rejection. As it stands,

the paper is rather a preliminary report on one single core, and more investigation is required before this study could be accepted for publication. The title itself does not reflect exactly the content of the paper, as the distinction between storm and tsunami beds is not discussed in the paper. The potential storm deposits are not (or very-poorly) described.

R1- Through the study of the TAH17-1 core, our objective was to identify marine submersion deposits in a general way, through their geochemical and sedimentological characteristics, without making a distinction between storm and tsunami deposits. We believe that a single core is sufficient to achieve this objective and we have presented the arguments for this. Concerning the dated events, we presented some hypotheses concerning their establishment based mostly on the comparison between our geochronological data and other available archives, and we supported that with some representative sedimentary characteristics for marine submersion deposits (i.e. rip-up clasts, sediment discontinuity...). The state of knowledge on marine submersion in Morocco remains very limited compared to neighbouring countries (Spain and Portugal). Even if we have not dealt with some desired aspects, such as the separation between tsunami and storm deposits for all the identified events, we present through our core the history of marine submersions during the last 4000 years on the Moroccan Atlantic coast, and as you said "The sediment core that is presented in your paper represents the most complete sequence of marine submersion events ever found on the coasts of Morocco". The title of the manuscript may be confusing for the readers, we will try to improve it for the next version.

Q2- In the introduction, more references are needed (see also my comment below). Previous references on marine submersion events on the coasts of Morocco are cited, but it would be good to add a short paragraph summarising the main conclusions of these papers.

R2- We will formulate a synthesis of these marine submersion deposits in Morocco and integrate it into the next version.

Q3- I was surprised that only one core (TAH17-1) is presented, Although at least one other core (TAH17-4, mentioned at line 20 page 7) was retrieved. Presenting other cores would allow evaluating the reproducibility of the results obtained for TAH17-1, together with giving information on the longitudinal/lateral continuity of the sedimentary units observed. Indeed, many tsunami deposits are characterised by longitudinal trends such as landward fining and thinning. Making conclusions from one core only is astrong limitation.

R3- Our objective is to identify marine submersions in a general way, coming from both phenomena (storms and tsunamis). If our objective was to separate between these type of deposits, we agree that a single core is not sufficient.

Q4- Methods: the geochemical analysis is based on data provided by a hand-held XRF device. The reliability of such equipment is questionable. As a first approach it gives a good idea of the composition of the sediment, but why not using a $\mu$-XRF lab device since you have good cores?

R4- The Handheld XRF (pXRF) has already been used in many studies (Chagué-Goff, 2017). If we question the reliability of the pXRF, we can do the same for the XRF core scanner ($\mu$XRF), since both devices work according to the same principle (X-ray fluorescence measurement). The results obtained with both tools are considered semi-quantitative, but still, a rapid way to know the geochemical composition of the sediment, with a very high resolution. For this purpose, a pXRF of the brand Niton Xl3t was made available to us by the Géosciences Montpellier laboratory to carry out geochemical measurements. Since marine submersion deposits are in general thicker than 1 cm in both cases (storm and tsunami deposits), which is why the use of pXRF with centimetric resolution was considered sufficient to identify all marine submersion deposits present in our core. Additionally, the device was calibrated with standards (provided by the manufacturer) and measurements were taken 150 seconds with the Soil mode, which is the most suitable mode for unconsolidated sediments.

Q5- Why merging results and discussion in section 4? Sections 4.1 and 4.2. are clearly"result" parts. Discussion really starts at page 7 (section 4.3). Please consider re-organising the manuscript so as to separate results and discussion.

R5- Your comment will be taken into consideration. The results and discussion will be re-organised as requested.

Q6- Description of the lithostratigraphy is too short and should include a description of the grain size distribution of individual subunits, and vertical trends of grain size.

R6- We will take into consideration your comment.

Q7- On figure 2, there's a sedimentary gap at depth 90-120 cm. It is thus difficult to have a good idea of the characteristics of unit B, which is a key unit, probably related to the 1755 tsunami. Is it due to sand percolation during core retrieval?

R7- The core drilling machine used allows the extraction of sediment metre by metre. During the extraction step, we noticed sediment compaction equivalent to 10 cm on the first meter, and 25 cm on the second. This is why we have a "gap" between 90-125 cm. There was no loss of sediment, but rather compaction. Concerning the characteristics of unit B, the study of this unit in details was difficult, because the unite was structureless (as you can see in fig. 2), except for some major characteristics that we have already mentioned in the article: the rip-up clasts and sediment discontinuity at the base.

Q8- In that case why not trying other cores? Why not trenching in order to access to this important unit?

R8- During our field mission, we recovered other cores. Laboratory analyses are ongoing for some sections (TAH17-4), but for others they have not yet started. The results will be published once the analyses are completed. We tried to make trenches in some coring points, but each time, the groundwater level was close to the surface (zero coring level). The sediment began to fluidize and flowed over the trench, making difficult

for us to collect samples, especially when you are working on centimetric resolution.

Q9- It's another severe limitation for the conclusions that are proposed. At the beginning of page 5, unit B is compared to other 1755 tsunami deposits in Portugal. This comparison could be enlarged to many other tsunami deposits in the world (i.e. marine sand beds as discontinuities in finer sediments), but the sedimentary setting in Portugal and Morocco is certainly different, so more arguments are needed.

R9- We all agreed that each site has its own sedimentary setting, but marine submersion deposits worldwide share the same sedimentary criteria, especially in estuarine and lagoon systems (as you mention it: "marine sand beds as discontinuities in finer sediments"). We focused on Portugal and Spain because these two countries share a common history of extreme marine events with Morocco, especially for tsunamis (i.e. Lisbon 1755 AD).

Q10- The XRF profiles seen on figure 4 should be described in more details (actually the section presenting these results is only 5-lines long).

R10- Your recommendation will be taken into account.

Q11- At page 9 (correlation of events with historical archives), we lack a discussion on the magnitude/intensity of the 1755 tsunami on the Moroccan coast. There's indeed a great debate on the impact of this event and the Moroccan piece of the puzzle is particularly important for deciphering between the different rupture scenarios that are proposed in the litterature (see papers by MA Baptista, PL Blanc, etc). The tsunami evidence at 1km from the shoreline provided by your new data is a key observations, but it is (too) briefly presented and discussed, and so under-exploited.

R11- Comparing our field results with the proposed models by MA Baptista, PL Blan... would be the ultimate goal we are trying to achieve through our project. but for that, we need to know the actual extent of our deposits in the field, and for that, it is necessary to retrieve more cores (by default several field missions), supported by geophysical

data (especially with "Ground penetrating radar" for our study area). Unfortunately, it requires great resources and a lot of time. we consider our study as a first step to achieve this ultimate goal.

Q12- In the description of the core, you distinguished E13 and E14 subunits, but suddenly in the discussion these 2 subunits become a single tsunami deposit because it fits better with ages! A better description of the E13/E14 section of the core is required before concluding if it's one or 2 different events.

R12- Our hypothesis to explain why E13 and E14 are considered as a single event are based mostly on geochronological data. The C14 dates for these two sandy layers are close, around 3200 BP, and some marine high energy deposits found on the Spanish coast are approximately the same age. We hypothesise (not conclude) that E13 and E14 may be deposited during a single event (around 3200 BP) through multiple waves.

Q13- By the way, English should be revised by a native.

R13- The English will be revised by a native as requested.

Q14- References: More references in the introduction would be welcome (e.g. at lines 23, 27, 34, 36). Please avoid auto-citation when you cite a single paper illustrating a large topic (e.g.at line 23: Chaumillon et al. 2017 deals with storms only, although it's an excellent review).

R14- Corrections will be made for this part.

Q15- Minor comments: p1-l33: not sure if a tsunami from Alboran Sea would impact the Atlantic coast of Morocco.

R15- The sentence will be corrected.

Q16- p1-l34: plunging (not diving).

R16- This has been corrected as requested.

Q17- p1-l36-38: the Cumbre Vieja case viewed from Ward & Day (2001) is highly controversial. Please add other references on this low-probability / high-magnitude scenario (e.g. Abadie et al., 2012).

R17- the reference will be added.

Q18- p2-l3: "This approach has been used on the Portuguese and Spanish coasts, especially to track past tsunamis."

R18- This sentence has been corrected.

Q19- p4-l36: I don't understand why unit B is 20 cm thick between 140 and 85 cm depth...

R19- on the core, unit B is 20 cm long. it's composed of 15 cm (between 140 and 125 cm) and 5 cm (between 90 and 85 cm). However, since there was no sediment loss during core extraction, we assume that unit B was thicker, probably 55 cm (space between 140 and 85 cm).

Q20- p5-l1-2: this sentence should be moved to discussion.

R20- The two lines will be moved into the discussion.

Q21- p5-l9-12: this section belongs to "methods".

R21- This part will be moved to the methods part, as requested.

Q22- p5-l20: silica (rather than silicone).

R22- The periodic table indicates the chemical element "Si" as Silicon. The word Silica is used for silicon dioxide ($SiO_2$).

Q23- p7: it would be good to remind the depth range of subunits E1, E13 and E14 in the text.

R23- Depth range for E1, E13 and E14 has been added as requested.

Q24- p7-l15: I'm not familiar with the use of such extrapolated ages, but is it really correct to present such an averaged date from two different techniques?

R24- Extrapolated ages using Cs-137 and Pb-210 has been used in previous studies (Costa et al., 2012; Pouzet et al., 2019). Maybe it is preferable to present all ages in the paper according to one notation system (i.e. BCE/CE system), we will do that for the next version of the manuscript.

Q25- p7-l23: The reservoir effect is not the only possible explanation for the difference observed between C-14 age on shell and wood. Where does they both come from in such a sedimentary setting? no transport prior to their present-day position?

R25- both samples were extracted from a silty mud level in the TAH17-4 core, which indicates a low energy environment and thereby reducing the probability of finding old remobilized materials.

Q26- p10-l13: I would not write "high-resolution". High-resolution studies rely on other techniques such as $\mu$-XRF, CT-scan, etc.

R26- We think that high-resolution, low-resolution depends on which proxy we are referring. For the geochemical proxy, we have a lower resolution with the pXRF compared to $\mu$-XRF. Otherwise, we have a high-resolution for the grain size analysis. However. we think that a paleoenvironmental study every cm on a long sedimentary archive still a high-resolution study.

Q27- Figures 4 and 5 could be merged together. Place subunits E1, E13 and E14 on figures 2 and 4.

R27- Your remarque will be taken into consideration. we will try improve figure 2 and figure 4.

Références:

Chagué-Goff, C., Szczuciński, W. and Shinozaki, T.: Applications of geochemistry in tsunami research: A review, Earth-Science Rev., 165, 203–244, doi:10.1016/j.earscirev.2016.12.003, 2017.

Costa, P. J. M., Andrade, C., Freitas, M. C. da C., Oliveira, M. A. and Jouanneau, J.-M. M.: Preliminary Results of Exoscopic Analysis of Quartz Grains Deposited by a Palaeotsunami in Salgados Lowland (Algarve, Portugal), J. Coast. Res., 2009(SI 56), 39–43, 2009.

Pouzet, P., Maanan, M., Sabine, S., Emmanuelle, A. and Robin, M.: Correlating three centuries of historical and geological data for the marine deposit reconstruction of two depositional environments of the French Atlantic coast, Mar. Geol., 407, 181–191, doi:10.1016/j.margeo.2018.10.014, 2019.

---

## Referee Comment (RC2) · Pedro Costa (Referee) · 28 Aug 2019

Dear authors, the work presented by Khalfaoui et al. focus on the reconstruction of storm and tsunami events in the Atlantic coast of Morocco. This is an interesting contribution on a very relevant topic for the definition of risk and return periods for the Atlantic coast of Morocco and broadly speaking for the Gulf of Cadiz.

Despite a considerable effort made by the authors the manuscript has many shortcommings and in some case can be considered poorly supported by the data. For me, one crucial aspect is related with a sentence on page 7 - "Based on our sedimentological and geochemical results, 14 sediment layers attributed to marine submersion events

have been distinguished in TAH17-1 core (Fig. 5). These layers present at least one of the following characteristics: - a well-defined sandy layer, interbedded in fine estuarine sediments; - an enrichment in Ca and Zr indicating a marine source of the sediment, and depletion in terrigenous elements (Rb); - the presence of sedimentation structures such as rip-up clasts and/or sediment discontinuity"

The problem with this sentence is the "at least" that clearly demonstrated a lighthearted approch to tsunami sedimentology. I am strongly opposed to that approach- To assign a deposit as a tsunami-related unit one should present cummulative data - thinning and fining inland, erosive or abrupt basal contact, coarser than under and overlying layers, changes in geochemical, plaeontological and compositional features, etc. There are abundant review papers on tsunami sedimentology that focus on the multiple and vast approaches that need to be conducted before jumping to conclusions. In this manuscrip by Khalfaoui et al. a potential good study case is not well-presented because of lack of compreenhsion on the care that must be taken when assigning a layer to an extreme marine inundations. Therefore, based on a single core it is highly speculative to identify 14 events and in many cases based on a single sedimentological criteria (which is contrary to what reference works state). I am aware of the work conducted and the different analysis the authors did however, the data is poorly presented and clearly this manuscript is not mature enough for publication. You need to

I tottally subscribe the detailed comments made by Reviewer Raphael Paris and I am not going to repeat his brilliant review.

Furthermore, I think the authors should take a step back, produce and present more data and them resubmit the manuscript. The work has potential but it is not ready for publication.

Therefore, I regret to say but I recommend this manuscript to be rejected. Kind regards Pedro JM Costa

2019-130, 2019.

---

## Author Comment (AC2) · 7 Oct 2019

The authors would like to thank Professor Pedro JM Costa for his valuable comments and suggestions. they will certainly contribute to a significant improvement of the article. However, we present some clarification and answers (R) to his questions (Q) in the following text:

Q1. Dear authors, the work presented by Khalfaoui et al. focus on the reconstruction of storm and tsunami events in the Atlantic coast of Morocco. This is an interesting contribution on a very relevant topic for the definition of risk and return periods for the Atlantic coast of Morocco and broadly speaking for the Gulf of Cadiz. Despite a consid-

erable effort made by the authors the manuscript has many shortcomings and in some case can be considered poorly supported by the data. For me, one crucial aspect is related with a sentence on page 7 - "Based on our sedimentological and geochemical results, 14 sediment layers attributed to marine submersion events have been distinguished in TAH17-1 core (Fig. 5). These layers present at least one of the following characteristics: - a well-defined sandy layer, interbedded in fine estuarine sediments; - an enrichment in Ca and Zr indicating a marine source of the sediment, and depletion in terrigenous elements (Rb); - the presence of sedimentation structures such as rip-up clasts and/or sediment discontinuity". The problem with this sentence is the "at least" that clearly demonstrated a lighthearted approach to tsunami sedimentology. I am strongly opposed to that approach- To assign a deposit as a tsunami-related unit one should present cumulative data - thinning and fining inland, erosive or abrupt basal contact, coarser than under and overlying layers, changes in geochemical, palaeontological and compositional features, etc. There are abundant review papers on tsunami sedimentology that focus on the multiple and vast approaches that need to be conducted before jumping to conclusions. In this manuscript by Khalfaoui et al. a potential good study case is not well-presented because of lack of comprehension on the care that must be taken when assigning a layer to an extreme marine inundations. Therefore, based on a single core it is highly speculative to identify 14 events and in many cases based on a single sedimentological criteria (which is contrary to what reference works state).

R1. This sentence needs to be corrected. in fact, all sandy layers identified in TAH17-1 core as marine submersion events have at least two distinctive criteria: (1) these deposits are coarser than under and overlaying layers; seconde, (2) they present high concentration of Ca and Zr, which indicate a marine source of the sediment. At this level, these two independent parameters (grain size + geochemistry) are sufficient to conclude that we have a marine submersion deposit (regardless of the type of event causing these deposits). These two criteria have been used to determine marine submersion deposits in previous studies (Raji et al., 2015; Avşar, 2019; Zhou et al., 2019).

Some sandy layers in the TAH17-1 exhibit more distinctive criteria, like the E1 event, with the presence of rip-up clasts and erosive basal contact with underlying unite. Perhaps it would be more interesting to present this part of the results through a table with all known sedimentological and geochemical criteria and to check for each identified level in this study whether these criteria are verified or not.

Q2: I am aware of the work conducted and the different analysis the authors did however, the data is poorly presented and clearly this manuscript is not mature enough for publication. You need to I totally subscribe the detailed comments made by Reviewer Raphael Paris and I am not going to repeat his brilliant review. Furthermore, I think the authors should take a step back, produce and present more data and them resubmit the manuscript. The work has potential but it is not ready for publication. Therefore, I regret to say but I recommend this manuscript to be rejected. Kind regards Pedro JM Costa

R2: Corrections and suggestions provided by Professeur Raphaël Paris are for sure important to improve the manuscript. We subscribe also for his detailed comments.

References

Avşar, U.: Sedimentary geochemical evidence of historical tsunamis in the Eastern Mediterranean from Ölüdeniz Lagoon, SW Turkey, J. Paleolimnol., 61(3), 373–385, doi:10.1007/s10933-018-00065-x, 2019. Raji, O., Dezileau, L., Von Grafenstein, U., Niazi, S., Snoussi, M. and Martinez, P.: Extreme sea events during the last millennium in the northeast of Morocco, Nat. Hazards Earth Syst. Sci., 15(2), 203–211, doi:10.5194/nhess-15-203-2015, 2015. Zhou, L., Yang, Y., Wang, Z., Jia, J., Mao, L. and Li, Z.: Investigating ENSO and WPWP modulated typhoon variability in the South China Sea during the mid – late Holocene using sedimentological evidence from southeastern Hainan Island , China, Mar. Geol., 416(July), 105987, doi:10.1016/j.margeo.2019.105987, 2019.
* * *
2019-130, 2019.